# Adenosine A_3_ Receptor (A_3_AR) Agonist for the Treatment of Bleomycin-Induced Lung Fibrosis in Mice

**DOI:** 10.3390/ijms232113300

**Published:** 2022-11-01

**Authors:** Silvia Sgambellone, Silvia Marri, Stefano Catarinicchia, Alessandro Pini, Dilip K. Tosh, Kenneth A. Jacobson, Emanuela Masini, Daniela Salvemini, Laura Lucarini

**Affiliations:** 1Department of Neuroscience, Psychology, Drug Research and Child Health (NEUROFARBA), Section of Pharmacology, University of Florence, Viale Gaetano Pieraccini, 6, 50139 Florence, Italy; 2Department of Experimental and Clinical Medicine, Section of Histology, University of Florence, Viale Gaetano Pieraccini, 6, 50139 Florence, Italy; 3Laboratory of Bioorganic Chemistry, National Institute of Diabetes & Digestive & Kidney Diseases, National Institutes of Health, 9000 Rockville Pike, Bethesda, MD 20892, USA; 4Pharmacology and Physiology, Saint Louis University, School of Medicine, 1402 South Grand Blvd, St. Louis, MO 63104, USA

**Keywords:** adenosine A_3_ receptor, inflammation, lung fibrosis, A_3_AR agonist, MRS5980

## Abstract

Adenosine receptors (ARs) are involved in the suppression and development of inflammatory and fibrotic conditions. Specifically, AR activation promotes differentiation of lung fibroblasts into myofibroblasts, typical of a fibrotic event. Pulmonary fibrosis is a severe disease characterized by inflammation and fibrosis of unknown etiology and lacking an effective treatment. The present investigation explored the action of MRS5980, a new, highly potent and selective A_3_AR agonist, in an established murine model of lung fibrosis. The effects of either vehicle or MRS5980 were studied in mice following intratracheal bleomycin administration. We evaluated the role of the A_3_AR agonist on lung stiffness, studying the airway resistance to inflation, oxidative stress (8-OHdG and MDA), inflammation, pro- and anti-inflammatory marker levels (IL-1β, IL-6, TNF-α, IL-10 and IL-17A) and fibrosis establishment, evaluating transforming growth factor (TGF)-β expression and α-smooth muscle actin (α-SMA) deposition in lungs. Bleomycin administration increased lung stiffness, TGF-β levels, α-SMA deposition, and inflammatory and oxidative stress markers. The treatment with MRS5980 attenuated all the analyzed functional, biochemical and histopathological markers in a dose-dependent manner. Our findings support the therapeutic potential of A_3_AR agonists in lung fibrosis by demonstrating reduced disease progression, as indicated by decreased inflammation, TGF-β expression and fibrotic remodeling.

## 1. Introduction

Lung fibrosis is a progressive disorder with a poor prognosis, characterized by interstitial fibrosis of the lung as a pathological response to chronic inflammation, which leads to epithelial cell damage, vascular exudation and leukocyte infiltration in the alveolar spaces [1,2]. Although its etiology remains unknown, the condition is characterized by an increase in fibroblast proliferation and their activation into myofibroblasts with a trans-differentiation of epithelial cells and a massive formation of collagen together with other matrix components, such as fibronectin and α-smooth muscle actin (α-SMA) [3]. This process leads to progressive airway stiffening, impairing breathing and eventually resulting in respiratory failure. Pulmonary fibrosis (PF) is a severe pulmonary disease of increasing prevalence and a high mortality rate, with median survival of only ~3 years [4]. Actually, there are only two therapies useful to treat this disease—i.e., pirfenidone, a fibroblast proliferation inhibitor, and nintedanib, a tyrosine kinase inhibitor—but, unfortunately, their efficacy is insufficient [5,6,7] and some patients still require lung transplantation [4]. Therefore, novel treatment approaches, based on first-in-class molecules and/or new therapeutic pathways, are desperately needed.

The endogenous nucleoside adenosine is released by metabolically active cells or generated through the hydrolysis of extracellular ATP by ectonucleotidases. During injury or stress conditions, released extracellular ATP is converted to adenosine, which induces many physiological effects through autocrine or paracrine activation of four subtypes of G-protein-coupled receptors, termed A_1_, A_2A_, A_2B_ and A_3_ adenosine receptors (ARs). Mouse and human lungs express all four receptors, which have roles in balancing pro- and anti-inflammatory pathways, inducing bronchoconstriction, and regulating pulmonary inflammation and airway remodeling, depending on the receptor subtypes [8]. ARs also promote lung fibroblast differentiation into myofibroblasts, which is a characteristic of the fibrotic event. This latter function suggests that ARs are potentially involved in pulmonary fibrotic progression, which is characterized by varying degrees of inflammation and fibrosis [9].

The role of A_3_AR in the lung is not clearly defined, and conflicting data indicate that further investigation of its pulmonary role is needed. It is known that the A_3_AR has a key role in regulating lung inflammation and airway remodeling; in fact, A_3_AR levels are elevated in patients with chronic lung disease and various pulmonary pathological conditions [8]. Moreover, A_3_AR activation using A_3_AR agonists in rodents induces mast cell degranulation, bronchoconstriction and mucus secretion [10]. Conversely, A_3_AR knock-out mice developed enhanced pulmonary inflammation with increased production of eosinophil-related chemokines and cytokines [8]. Finally, in a bleomycin model of pulmonary inflammation and fibrosis, the A_3_AR induced an anti-inflammatory process by regulating the production of inflammatory and fibrotic mediators [8,11].

The novel and highly selective A_3_AR agonist, MRS5980, is representative of a new generation of A_3_AR agonist that are effective in several animal experimental models of neuropathic pain [12,13,14,15].

Previous studies showed that this compound effectively controls chronic but not acute pain through its anti-inflammatory action [16]; in addition, the analysis of the metabolic mapping of this compound highlighted that MRS5980 is extremely safe compared with other A_3_AR agonists, and it does not develop tolerance [16]. These findings suggested a possible use of MRS5980 for the treatment of other chronic diseases such as, for example, pulmonary fibrosis.

Starting from the evidence showing that A_3_AR has an anti-inflammatory effect in pulmonary fibrosis and that the highly selective A_3_AR agonist MRS5980 is extremely effective in chronic neuropathic pain, we evaluated the ability of MRS5980 to control lung inflammation and fibrosis in an established mouse model of bleomycin-induced pulmonary fibrosis.

## 2. Results

### 2.1. A_3_AR Agonist Effects in Ameliorating Lung Function and Pulmonary Architecture

The experimental protocol to obtain bleomycin-induced lung fibrosis in mice and the administered treatments (vehicle, MRS5980 1 mg/kg and 3 mg/kg) is shown in Figure 1A. Following intratracheal bleomycin stimulation, there was a significant increase in lung weight and airway stiffness resulting in a distinct elevation of the pressure at airway opening (PAO) [2]. In the present experimental series, we evaluated PAO in naïve mice, bleomycin-stimulated mice and in bleomycin plus MRS5980-treated mice. Figure 1B shows that bleomycin stimulation increased PAO from 17.8 ± 0.5 to 31.9 ± 1.4 mm, while the administration of MRS5980 at two different doses, 1 and 3 mg/kg/day, b.i.d., for 21 days attenuated the bleomycin effects on PAO (28.0 ± 0.8 and 25.1 ± 1.2 mm, respectively). The histological panel in Figure 1C shows representative images of the lung parenchyma of each group, stained by hematoxylin/eosin. In the naïve group, normal lung architecture is maintained, while an altered lung architecture is present in the vehicle group. The bleomycin-induced changes are attenuated by MRS5980 treatment in a dose-dependent manner.

It is well known that an increase in lung weight, induced by intratracheal injection of bleomycin, is an indicator of lung injury [17]; Figure 1D confirmed data reported in the literature, showing an increase in lung weight in the vehicle group (from 82.3 ± 1.6 to 111.2 ± 3.7 mg). However, the administration of MRS5980, especially at the highest dose of 3 mg/kg, improved the damage in a statistically significant manner (107.1 ± 1.9 and 92.5 ± 2.8 mg, respectively).

Figure 2A shows the hydroxyproline content in lung tissue homogenates. In animals, hydroxyproline is found almost entirely in collagen, and its content in tissue hydrolysates is a direct measure of the amount of collagen present, hence proportional to fibrosis. In control mice (naïve), the basal hydroxyproline content was 3.3 ± 0.49 ng/μg of protein, while with vehicle it was 4.21 ± 0.23 ng/μg of protein. However, MRS5980 treatment clearly attenuated this effect in a dose-dependent manner (3.75 ± 0.22 ng/μg and 3.11 ± 0.14 ng/μg, at 1 and 3 mg/kg, respectively), resulting in a reduction in the hydroxyproline content by 24.5% in the 1 mg/kg group and 59.3% in the 3 mg/kg group in comparison with the vehicle group. These results strengthen the functional and morphological findings on lung fibrosis.

The results shown in Figure 2B,C demonstrate that bleomycin administration amplified collagen deposition in interstitial lung spaces with extensive damage of the alveolar structure. The treatment with MRS5980 decreased these pathological changes, and the morphological lesions were significantly reduced.

To obtain data on the bronchial smooth muscle layer status, we measured the thickness of the muscle layer using a morphometrical analysis in hematoxylin/eosin-stained slides. The thickness of the airway smooth muscle layer was increased in mice treated with bleomycin, as expected, while MRS5980 treatment decreased this damage dose-dependently (Figure 3A,B).

PAS-stained preparations were used to evaluate bronchial mucosa goblet cell hyperplasia. The percentage of PAS-positive goblet cells over total bronchial epithelial cells increased in bleomycin-stimulated mice, while in MRS5980-treated animals, this marker of mucosal damage was significantly reduced in a dose-dependent manner (Figure 3C,D).

### 2.2. TGF-β Signaling Pathway Assessment and Evaluation of Fibroblasts Activation

An increased TGF-β expression contributes to the formation and expansion of pulmonary fibrosis [18]. We confirmed that bleomycin administration produced a large increase in plasma TGF-β levels (from 23.3 ± 7.5 to 131.8 ± 9.7 pg/mL). The treatment with MRS5980 (1 and 3 mg/kg/day) reduced this increase (89.0 ± 5.4 and 69.6 ± 10.6 pg/mL, respectively) (Figure 4A). At the highest dose (3 mg/kg), TGF-β levels decreased, approaching the naïve value. It is well known that TGF-β signaling regulates the expression of α-SMA, a marker of fibroblast activation and myofibroblast differentiation [6]. We evaluated the expression of α-SMA in lungs with an immunofluorescence analysis, highlighting a large increase in these levels in the vehicle group; meanwhile, in MRS5980-treated animals, α-SMA levels were considerably reduced. This result indicates that the A_3_AR-selective agonist reduces fibroblast activation and myofibroblast differentiation after bleomycin injection (Figure 4B,C).

### 2.3. Evaluation of Inflammation and Oxidative Stress Parameters

Chronic inflammation may lead to fibrotic progression in animal models of fibrosis, although the contribution of inflammation to fibrotic development is still unclear in the clinical disease manifestation. We evaluated the late inflammatory response to bleomycin after 21 days by measuring the following cytokines IL-1β, IL-6, IL-17A and TNF-α in mouse plasma samples (Figure 5A–D). Bleomycin stimulation consistently increased their levels (IL-1β from 0.5 ± 0.2 to 1.2 ± 0.5 pg/mL; IL-6 from 1.4 ± 0.3 to 6.3 ± 3.2 pg/mL; IL-17A from 0.6 ± 0.1 to 1.3 ± 0.2 pg/mL and TNF-α from 1.7 ± 0.3 to 2.7 ± 0.2 pg/mL), confirming a pivotal role of these pro-inflammatory cytokines in the development of pulmonary fibrosis as published by She and colleagues [19]. Strikingly, treatment with MRS5980 (1 and 3 mg/kg/day) dose-dependently reduced these increases (IL-1β: 0.4 ± 0.2 and 0.3 ± 0.2 pg/mL; IL-6: 2.3 ± 1.3 and 1.3 ± 0.5 pg/mL; IL-17A: 0.9 ± 0.2 and 0.7 ± 0.2 pg/mL and TNF-α: 1.5 ± 0.3 and 1.2 ± 0.4 pg/mL, respectively).

Moreover, bleomycin treatment significantly decreased the levels of regulatory cytokine IL-10 (from 6.5 ± 2.3 to 1.7 ± 0.7 pg/mL), while MRS5980 administration showed a trend toward increased IL-10 levels in a dose-dependent manner (3.1 ± 1.0 and 4.7 ± 3.0 pg/mL, respectively), even if it was not statistically significant (Figure 5E). Overall, these results suggest that treatment with the A_3_AR selective agonist reduced bleomycin-induced lung fibrosis involving the immune response through the inhibition of pro-inflammatory cytokines.

Furthermore, it is well known that mast cells appear to have a critical role in the development of inflammation and pulmonary fibrosis, and therapeutic blockade of mast cell degranulation should attenuate these processes [20]. Evidence is reported on the effects of adenosine on mast cells [10].

To address this point, we evaluated mast cell degranulation in the lung following the administration of MRS5980. A significant decrease in mast cell optical density upon staining with Astra Blue, for the massive discharge of mast cell granules, has been reported in the lungs of bleomycin plus vehicle treated mice, while MRS5980 3 mg/kg treatment reduced mast cell degranulation (Figure 6).

Finally, measurements of 8-OH*d*G (Figure 7A)—a marker of oxidative DNA damage—indicated that bleomycin administration significantly increased 8-OH*d*G level (from 1.2 ± 0.16 to 4.52 ± 0.52 pg/μg of DNA), while MRS5980 treatment (1 and 3 mg/kg/day) caused a dose-dependent reduction of this parameter (0.90 ± 0.27 pg/µg and 0.77 ± 0.42 pg/µg, respectively), suggesting that A_3_AR activation contributes to the control of oxidative lung damage. To confirm the results on oxidative stress effects, we evaluated the production of malondialdehyde (MDA), as thiobarbituric acid reactive substance (TBARS), end-products of cell membrane lipid peroxidation by ROS, a reliable marker of oxidative tissue injury (Figure 7B). The production of TBARS (MDA equivalents) was markedly increased in the vehicle group in comparison with naïve animals (from 24.47 ± 2.07 to 76.00 ± 4.70 ng/mg of protein). As shown in Figure 7B, MRS5980 treatment significantly reduced TBARS production at the highest dose (1 mg/kg 59.61 ± 8.18 and 3 mg/kg 39.30 ± 2.95 ng/mg of protein).

## 3. Discussion

Pulmonary fibrosis is a pathological chronic condition that negatively affects a potentially increasing number of people in the world. Although several therapeutic strategies are used clinically to treat this disease, none of the available drugs significantly reduce patient mortality [21]. Previous studies demonstrated the involvement of the adenosinergic system in pulmonary fibrosis development. In fact, all four ARs are expressed in the lungs of humans and mice, with different roles—in particular, A_3_AR was an important regulator of lung inflammation and airway remodeling [9].

A new, highly potent and selective A_3_AR agonist, MRS5980, demonstrated to be effective in several experimental models. In fact, this compound reduced secondary events and improved neurocognitive functions following traumatic brain injury [22]. It mediated the relief of acute visceral pain in rats [23]; moreover, it prevented and reversed chemotherapy-induced neurotoxicities in mice [24] and, finally, it was proved to reverse neuropathic pain [25]. Thus, since MRS5980 is an effective therapeutic intervention in other rodent models, we tested its efficacy in an animal model of lung fibrosis in the hope of finding a new therapy, as the involvement of the adenosinergic system in the development of this pathology has been widely demonstrated (9).

The A_3_AR agonist MRS5980 was studied in the present research to investigate its effects in reducing the functional and structural characteristics of bleomycin-induced lung fibrosis in mice. Various approaches have been used to induce pulmonary fibrosis in experimental animals, which facilitate the probing of mediators and processes likely involved in human lung fibrosis, which is a progressive and fatal disease. Intratracheal bleomycin injection is the most current and well-characterized model that closely mimics human pulmonary fibrosis [26].

Our research was carried out in C57BL/6 mice, a strain susceptible to bleomycin-induced fibrosis [27]; the intratracheal injection of bleomycin causes a strong inflammatory response within the first week, followed by the development of fibrosis. Indeed, as already demonstrated [28,29], bleomycin stimulation caused a significant elevation in airway stiffness, as indicated by an increase in PAO in the fibrotic vehicle group compared with naïve mice, as well as alteration of the lung architecture in the vehicle group; these parameters were restored dose-dependently by daily MRS5980 treatments.

Histological parameters of adverse airway remodeling were evaluated by measuring bronchial mucosa goblet cell hyperplasia and the thickness of the smooth muscle layer. The percentage of PAS-positive goblet cells over total bronchial epithelial cells increased in bleomycin-stimulated mice, while in MRS5980-treated animals, this marker of mucosal damage was significantly reduced.

Moreover, bleomycin injection increased the thickness of the airway smooth muscle layer, as expected, while MRS5980 treatment decreased this damage in a dose-dependent manner. These findings confirmed data reported in literature [9] and suggested that this newly potent A_3_AR agonist was able to ameliorate the bronchial remodeling.

Furthermore, we evaluated the inflammatory response to lung injury induced by bleomycin, in plasma samples of mice treated with vehicle or MRS5980. The results showed that the treatment with this compound significantly reduced the levels of pro-inflammatory cytokines such as IL-1β, IL-6, IL-17A and TNF-α, as well as lung levels of 8-OH*d*G and MDA, markers of oxidative stress. Moreover, MRS5980 treatment tended to increase IL-10 levels, suggesting an involvement of this regulatory cytokine. In addition, it is known that the A_3_AR plays an important role in adenosine-mediated murine lung mast cell degranulation [30]. The functional role of adenosine A_3_AR in mast cell activation was studied using adenosine A_3_AR and mast cell knock-out mice. Mast cell degranulation depends on the activation of this receptor, suggesting that the A_3_AR-dependent airway inflammation is a consequence of adenosine exposure [10].

Conversely, in a bleomycin model of pulmonary inflammation and fibrosis, the A_3_AR displayed anti-inflammatory effects, regulating the production of the mediators involved in fibrosis [8,11]. Our results are in agreement with these last observations; in fact, the treatment with MRS5980 at a dose of 3 mg/kg reduced mast cell degranulation in the lungs. Noam and colleagues demonstrated an inverse relationship between the A_3_AR expression level and the expression of tissue remodeling target genes; in fact, the activation of A_3_AR suppresses its own expression both at protein and transcriptional levels [31]. Thus, we can speculate that the administration of MRS5980 could prime this feedback mechanism.

Interestingly, this compound, particularly at the highest dose, has proven to be effective in diminishing the inflammatory response, oxidative stress damage induced by production of free radicals and mast cell degranulation.

TGF-β and its signaling pathway are known to play a key role in the development of pulmonary fibrosis [32], and treatments aimed at controlling the levels of this cytokine appear to be effective in reducing the fibrotic process, modulating fibroblast activation and, consequently, the progression of the disease [33]. These findings are perfectly in agreement with the results of the present research; in fact, TGF-β levels are significantly increased after bleomycin stimulation, and the treatment with MRS5980 at the highest dose drastically reduced plasma levels of TGF-β, dampening its transduction pathway. This result was confirmed by the reduced deposition of α-SMA, indicating a prevention of fibroblast activation and myofibroblast differentiation. Furthermore, MRS5980 treatment reduced collagen accumulation in interstitial lung spaces, as shown by picrosirius red staining, and hydroxyproline content in the lungs, counteracting the extracellular matrix deposition and fibrosis development. This result suggested that A_3_AR signaling could be an important mechanism to link inflammation to progressive pulmonary fibrosis.

Taken together, our results are consistent with previous findings indicating that A_3_AR possesses anti-inflammatory influences [8] preventing the fibrotic process. Thus, our findings strengthen the notion that MRS5980 could be a potentially useful drug for the therapy of chronic inflammatory and fibrotic lung diseases. Indeed, the decrease in inflammation and the oxidative stress response has a significative impact on fibroblast activation and myofibroblast differentiation, inhibiting the progression of airway remodeling and the subsequent development of fibrosis.

### Study limitations

Our study exhibits some limitations; actually, despite the well-established model in mice, this experimental protocol has some constraints regarding the comprehension of human pulmonary fibrosis progression. In fact, bleomycin injection induces changes that mimic the fibrosis process, but it does not fit perfectly with every stage of development of the human disease. A deep investigation of MRS5980’s mechanism of action could be interesting to provide further information on the adenosine signaling pathway in the control of pulmonary fibrosis development.

## 4. Materials and Methods

### 4.1. Drugs and Reagents

The compound MRS5980 (1*S*,2*R*,3*S*,4*R*,5*S*)-4-(2-((5-chlorothiophen-2-yl)ethynyl)-6-(methylamino)-9H-purin-9-yl)-2,3-dihydroxy-*N*-methyl-bicyclo [3.1.0]hexane-1-carboxamide (>3000-fold selectivity compared with A_1_AR, A_2A_AR or A_2B_AR) [34] was dissolved in a solution of 3% of dimethyl sulfoxide (DMSO, Sigma-Aldrich, St. Louis, MO, USA) in saline (0.9% NaCl), which was also used as vehicle.

Bleomycin (Merck-Millipore, Burlington, MA, USA) was used to induce lung fibrosis in the mouse model (see below). The drug doses and administration frequency were selected based on previous reports [2,25,35]. MRS5980 (1 mg/kg and 3 mg/kg body weight) and vehicle were administered intraperitoneally (i.p.) twice a day for 21 days.

### 4.2. Animals

Male C57BL/6 mice aged 8–9 weeks and weighing ~25–30 g were purchased from Charles River (Sant’Angelo Lodigiano LO, Italy). All animals were provided with a standard chow diet and water ad libitum, and were housed at 22 °C with a cycle of 12 h light/12 h dark for at least 48 h prior to the experiments. The study protocol complied with the European Economic Community (86/609/CEE) and the Declaration of Helsinki guidelines on animal experimentation and received approval from the University of Florence (Florence, Italy) Animal Ethical and Care Committee and the Italian Health Ministry (Authorization n° 874/2017-PR). Experiments were approved and performed following the ARRIVE guidelines at the Centre for Laboratory Animal Housing and Experimentation (CeSAL), University of Florence [36].

### 4.3. Surgery and Treatments

C57BL/6 mice (40 total) were anesthetized at day 1, using zolazepam/tiletamine, 50/50 mg/mL (Zoletil, Virbac Srl, Milan, Italy), at a dose of 50 µg/g body weight dissolved in 100 µL of saline i.p. [29]. An incision was made along the line of the neck and the trachea was exposed. An injection of bleomycin solution or saline was administered through the gap between the two tracheal cartilage rings. Thirty mice were stimulated with a single bleomycin dose (intratracheal injection, 0.05 IU diluted in 50 µL of saline), while the other 10 mice were treated similarly by intratracheal injection with 50 µL of saline (referred to as non-fibrotic negative controls, naïve). As reported in Figure 1A, starting from day 0, the bleomycin-stimulated mice, 10 per group, received two doses daily of 100 μL of the compound MRS5980 at different concentrations (1 and 3 mg/kg b.w.) by intraperitoneal (i.p.) administration, for the 21 days following bleomycin dosing. These are referred to as MRS5980-treated groups. Similarly, starting from day 0, the other ten mice were treated twice a day with vehicle (3% DMSO dissolved in saline) and referred to as fibrotic positive controls (vehicle). The mouse body weight was measured daily during 21 days of treatment in order to exclude any toxic effects of the drug treatment [29].

### 4.4. Functional Assay of Fibrosis

After 21 days of treatment, all mice were subjected to the measurement of airway resistance to inflation, a functional parameter indicative of fibrosis-induced lung stiffness. The measurement was performed using mechanical ventilation method of a constant volume and respiration rate per min [37,38]. Briefly, a 22-gauge cannula (Venflon 2; Viggo Spectramed, Windlesham, UK, 0.8 mm diameter) was introduced into the trachea of each anesthetized mouse. The mouse ventilation was controlled using a small-animal respirator (Ugo Basile, Comerio, Italy) adjusted to provide a tidal volume of 0.8 mL with a respiratory rate of 20 strokes/min. A high-sensitivity P75 type 379 pressure transducer (Harvard Apparatus Inc., Holliston, MA, USA), with settings of gain 1 and chart speed 25 mm/s, and a recording polygraph (Harvard Apparatus Inc. Edenbridge, UK) with settings of gain 1, chart speed 25 mm/s, were used to register the changes in lung resistance to inflation, i.e., the PAO. Changes in lung resistance to inflation, registered on the polygraph for at least 3 min and expressed as mm, were determined for 40 or more consecutive tracings of respiratory strokes before averaging [29].

### 4.5. Blood Withdrawal and Lung Tissue Sampling

After functional measurements of parameters for fibrosis-induced lung stiffness, the animals were sacrificed using anesthetic drugs at a lethal dose. Blood sampling was performed by cardiac puncture, and the plasma samples were obtained by blood centrifugation at 2000× *g* for 10 min at 4 °C. The entire left lung of each animal was removed and fixed for histological analysis using 4% paraformaldehyde in PBS. The right lung was weighed, quickly frozen and stored at −80 °C. For biochemical analysis, the lung samples were thawed at 4 °C, homogenized on ice in RIPA buffer and then centrifuged for 30 min at 10,000× *g* at 4 °C, unless otherwise indicated. The homogenized supernatants were stored for the following biochemical determinations.

### 4.6. Histology and Assessment of Collagen Deposition, Goblet Cell Hyperplasia, Smooth Muscle Layer Thickness and Mast Cell Degranulation

The paraffin-embedded lung samples were cut into 6 μm thick histological sections. All sections were stained in a single session to minimize artefactual inconsistencies during the staining process. A light microscope equipped with objectives with different magnifications and connected to a digital camera was used to record photomicrographs of the histological slides in a random fashion, with computer-aided densitometry to quantitatively assess the stained sections. Optical density (OD) and surface area were measured using the free-share ImageJ 1.33 image analysis program [39]. For each measured parameter, values are expressed as mean ± S.E.M. of the OD measurements (arbitrary units) using five images from each individual mouse in each experimental group (tested blind).

Lung collagen deposition was evaluated with picrosirius red staining. The sections were stained in 0.1% Direct Red 80/Sirius Red F3B (Sigma-Aldrich) in saturated picric acid at room temperature for 1 h; the sections were successively treated with 0.5% acetic acid prior to dehydration, clearing and mounting [40,41]. The degree of picrosirius red staining of collagen fibers was determined after selection of an appropriate threshold to eliminate aerial air spaces and bronchial/alveolar epithelium [29]. Furthermore, lung tissue sections were stained with hematoxylin/eosin and with periodic acid-Schiff (PAS) in order to quantify the morphometric parameters of smooth muscle layer thickness and bronchial goblet cell number, respectively, both strong indicators of airway remodeling. Moreover, to evaluate mast cell degranulation, we used Astra Blue staining. The analysis of mast cell degranulation was performed on lung sections fixed in Mota solution and successively stained with Astra Blue staining to evaluate mast-cell granule secretion. The histological evaluations were carried out on digital micrographs of the microscopic fields using the ImageJ 1.33 free-share image analysis software [38].

Microphotographs of small-sized bronchi were randomly recorded digitally. The bronchial smooth muscle layer thickness was measured in the digitized images using the ImageJ 1.33 image analysis software. Total bronchial epithelial cells and PAS-stained goblet cells in the bronchial cross-sections were counted, and the goblet cell percentage was calculated [29].

### 4.7. Hydroxyproline Assay

Lung tissue fibrosis was also quantified using a hydroxyproline assay kit according to the manufacturer’s protocol (BioVision, Milpitas, CA, USA). Briefly, the frozen lung tissue was lyophilized for 48 h and then thoroughly homogenized in distilled water. The samples were mixed with sodium hydroxide and hydrolyzed by autoclaving at 120° for 60 min in sealed polypropylene tubes. The samples were mixed with 10 N HCl and centrifuged at 10,000× *g* for 5 min. The oxidation reagent (chloramine T and oxidation buffer) was added to the hydrolysate and allowed to react for 20 min at room temperature. The absorbance of the mixture was read spectrophotometrically at 560 nm [42]. The results were calculated from a standard curve based on hydroxyproline solution.

### 4.8. Determination of α-SMA Deposition

Immunofluorescence was analyzed as previously reported [42]. Briefly, lung sections were deparaffinized and boiled for 10 min in sodium citrate buffer (10 mM, pH 6.0, purchased from Bio-Optica, Milan, Italy) for antigen retrieval. The sections were then sequentially immune-stained with rabbit monoclonal anti-α-SMA antibody (1:200; Abcam, Cambridge, UK) and goat anti-rabbit Alexa Fluor 568-conjugated IgG (1:300; Invitrogen, San Diego, CA, USA). Sections in which non-immune rabbit serum was substituted for the primary antibodies were taken as negative controls. The sections were counterstained with 4′,6-diamidino-2-phenylindole (DAPI), and representative images were acquired by an Olympus BX63 microscope (Milan, Italy) equipped with Olympus CellSens Dimension Imaging Software version 1.6. ImageJ software was used to quantify α-SMA expression in digitized images, by densitometric analysis of the intensity of the fluorescence signal. Each sample was analyzed in twenty regions of interest. Values are expressed as mean ± S.E.M. of the OD measurements (arbitrary units) of individual mice from the various experimental groups.

### 4.9. Cytokine Measurements

Levels of the cytokines interleukin-1β (IL-1β), interleukin-6 (IL-6), interleukin-17A (IL-17A), interleukin-10 (IL-10) and tumor necrosis factor-α (TNF-α) were quantitatively determined using an EDM bead-based multiplex immunoassay, following the manufacturer’s protocol (Millipore Corp., Billerica, MA, USA). In brief, neat plasma samples were added to antibody-conjugated beads directed against the abovementioned cytokines in a 96-well filter plate and incubated for 30 min. The plate was then washed, and the wells treated with biotinylated anti-cytokine antibody solution prior to overnight incubation. The plate was then washed, and streptavidin-conjugated PE was added. After a final wash, each well was treated with assay buffer and subjected to analysis using the Bio-plex 200 system (Bio-Rad, Milan, Italy). Standard curves were derived from multiple concentrations of standards of the various cytokines using the same protocol as for the plasma samples [43].

The levels of TGF-β, the major profibrotic cytokine involved in fibroblast activation, were quantified and expressed as pg/mL in plasma aliquots (100 μL) using a TGF-β_1_ mouse ELISA kit (ThermoFisher Scientific, Monza, Italy), following the manufacturer’s protocol.

### 4.10. Determination of 8-Hydroxy-2-Deoxyguanosine

Frozen lung samples were allowed to equilibrate to room temperature, and cellular DNA was isolated as previously reported [42] with minor modifications. Briefly, lung samples were homogenized in 1 mL of 10 mM PBS, pH 7.4; sonicated on ice for 1 min; and then treated with 1 mL of 10 mmol/L Tris-HCl buffer, pH 8, containing 10 mmol/L EDTA, 10 mmol/L NaCl and 0.5% SDS. The suspension was incubated for 1 h at 37 °C in the presence of 20 µg/mL RNase 1 (Sigma-Aldrich, Saint Louis, MO, USA) and then overnight at 37 °C under argon with 100 µg/mL proteinase K (Sigma-Aldrich). The mixture was extracted with a mixture of chloroform:isoamyl alcohol (10 volumes:2 volumes). DNA in the aqueous phase was precipitated upon addition of 0.2 volumes of 10 mmol/L ammonium acetate; solubilized in 200 µL of 20 mmol/L acetate buffer, pH 5.3; and denatured at 90 °C for 3 min. The extract was then supplemented with 10 IU of P1 nuclease (Sigma-Aldrich) in 10 µL and incubated for 1 h at 37 °C with 5 IU of alkaline phosphatase (Sigma-Aldrich) in 0.4 mol/L phosphate buffer, pH 8.8. The mixture was filtered using an Amicon Micropure-EZ filter (Merck-Millipore), and 50 µL of each sample was used for 8-hydroxy-2-deoxyguanosine (8-OH*d*G) determination using an ELISA kit (JalCA, Shizuoka, Japan), following the manufacturer’s protocol. The absorbance of the chromogenic product was measured at 450 nm and converted to units of ng/mg of DNA using a standard curve based on an 8-OH*d*G solution. The values are expressed as ng 8-OH*d*G/ng total DNA.

### 4.11. Malondialdehyde (MDA) Determination

Malondialdehyde (MDA), an end-product of cell membrane lipids peroxidation caused by oxygen derived free radicals, is considered a reliable marker of oxidative stress damage. The measurement of the chromogen was obtained from reaction of thiobarbituric acid reactive substance (TBARS) with 2-thiobarbituric acid [2]. Briefly, 50 mg of tissue were homogenized with 0.5 mL of Tris HCl buffer (50 mmol/L) containing 180 mM KCl and 10 mM EDTA, pH 7.4. Successively, the lung tissue pellets were dissolved in 0.5 mL of 2-thiobarbituric acid (1% *w/v*) in 50 mM NaOH and 0.5 mL of HCl (25% *w/v* in water). The samples were boiled in water for 10 min and, after cooling, the chromogen was extracted in 3 mL of 1-butanol. The organic phase was separated by centrifugation at 2000× *g* for 10 min. The absorbance of the organic phase was read at 532 nm wavelength. The values are expressed as nanograms of TBARS (MDA equivalents) per mg of proteins, using a standard curve of 1,1,3,3-tetramethoxypropane [2].

### 4.12. Statistical Analysis

Data were reported as mean ± S.E.M. of individual average measures of the different animals per group, for each assay. Significance of inter-group differences was evaluated by one-way ANOVA followed by Bonferroni post hoc test for multiple comparisons. Calculations were made using Prism 6 statistical software (GraphPad Software, Inc., San Diego, CA, USA), and significant differences corresponded to a probability value *p* < 0.05.

## 5. Conclusions

In conclusion, the present study highlights the importance of the correlation among A_3_AR signaling pathway, inflammation and pulmonary fibrosis progression. Several studies demonstrated that A_3_AR agonists are effective in controlling inflammation, but no evidence is known about the effects of these compounds on lung fibrosis. 

We can state that this study contributes to a better comprehension of the involvement of adenosine signaling pathway in lung disease, and in particular, the role of A3AR agonists in an animal model of lung fibrosis; we can hypothesize that MRS5980 or other A3AR selective agonists could represent a potential therapeutic strategy for the treatment of patients with chronic lung inflammation and fibrosis development, even if several aspects require further investigation. 

## Figures and Tables

**Figure 1 ijms-23-13300-f001:**
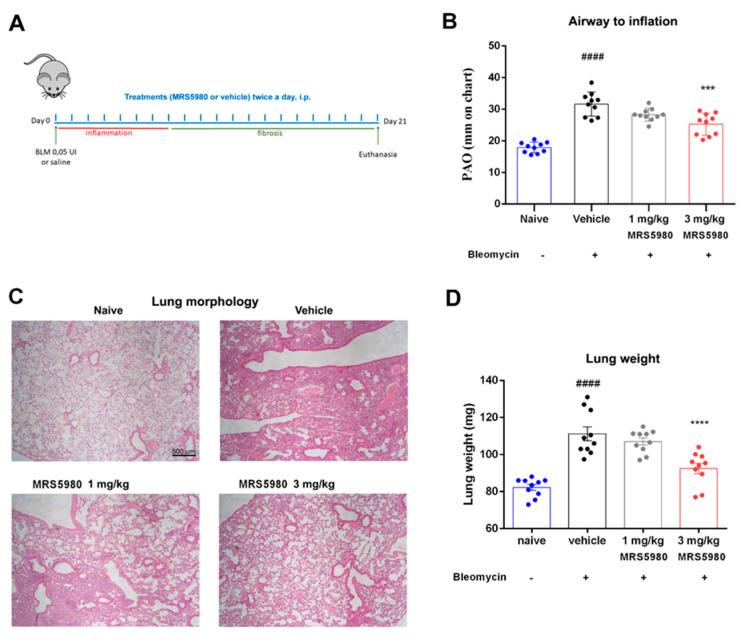
(**A**) Graphical scheme indicating the experimental protocol and treatments administered (BLM = bleomycin). (**B**) Lung resistance to airflow measured through the calculation of pressure at airway opening (PAO). (**C**) Histopathological assessment of lung parenchyma by hematoxylin and eosin staining of the different experimental groups (magnification 4×). (**D**) Bar graph showing the quantification of lung weight as a marker of lung injury in each experimental group. Data are mean ± S.E.M for n = 10 mice per group; #### *p* < 0.0001 vs. naïve; *** *p* < 0.001 and **** *p* < 0.0001 vs. vehicle.

**Figure 2 ijms-23-13300-f002:**
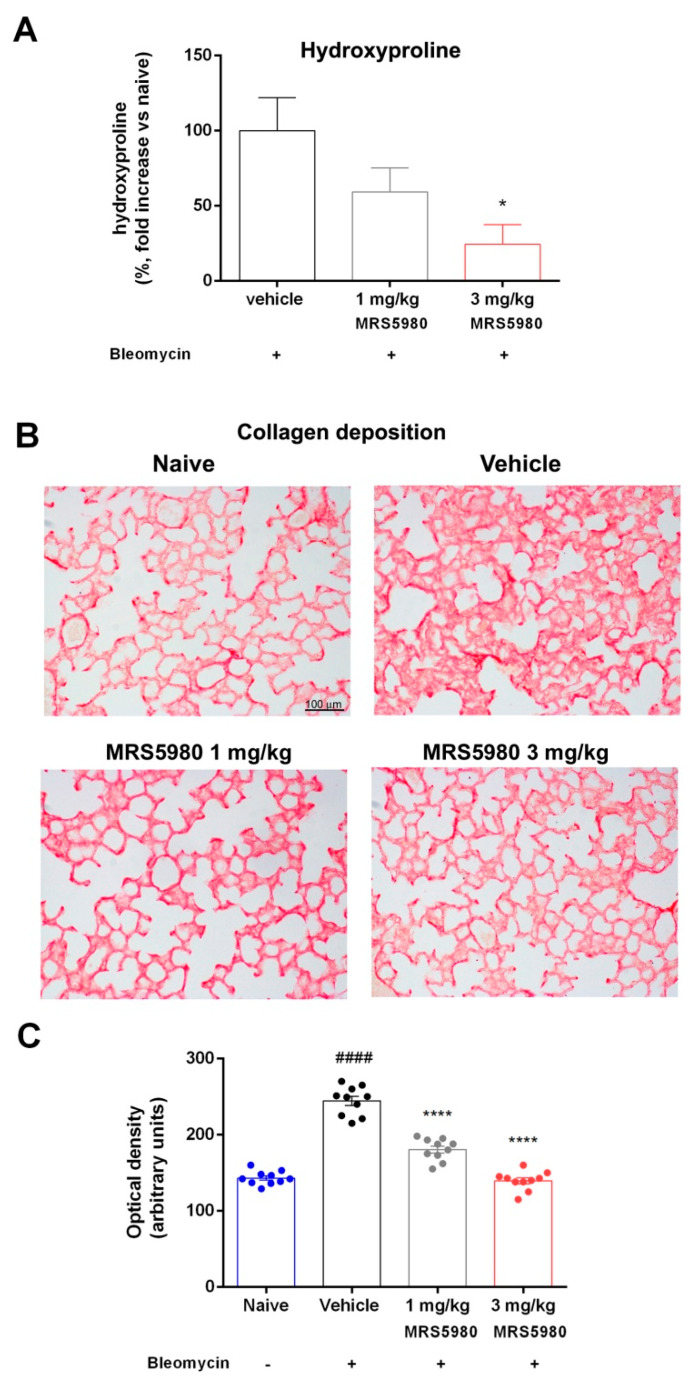
(**A**) Percentage decrease in hydroxyproline content (fold increase versus naïve), in comparison with vehicle, in lung homogenates. (**B**) Histopathological evaluation of collagen deposition by picrosirius red staining lung analysis (magnification 20×). (**C**) In the bar graph is reported a semi-quantitative measure obtained by computer-aided densitometry analysis. Data are mean ± S.E.M for n = 10 mice per group; #### *p* < 0.0001 vs. naïve; * *p* < 0.05 and **** *p* < 0.0001 vs. vehicle.

**Figure 3 ijms-23-13300-f003:**
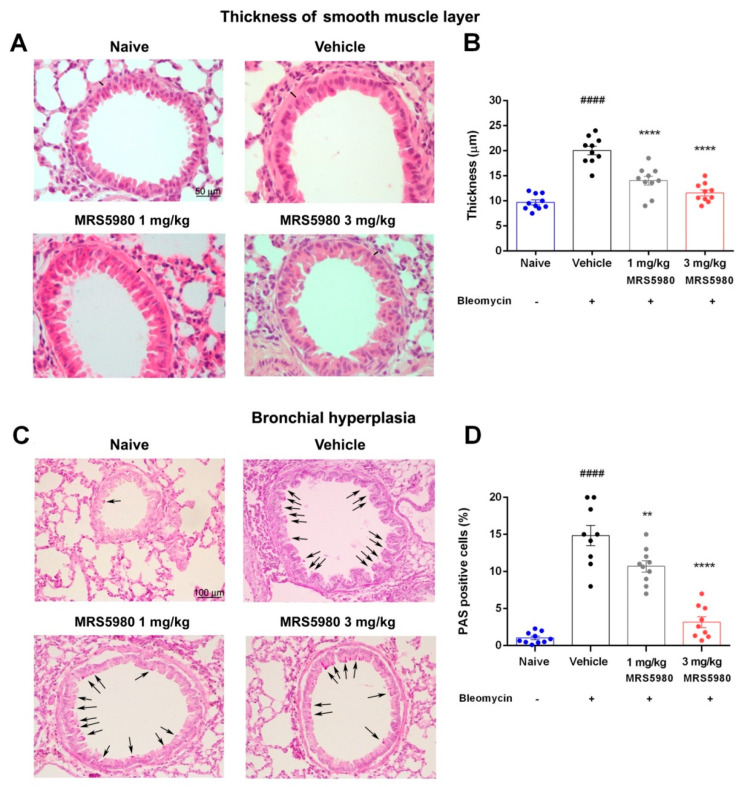
(**A**) Hematoxylin and eosin staining for the histopathological evaluation of the smooth muscle layer thickness in the airways at 40× magnification (black bars). (**B**) Bar graph shows the muscular thickness in the different experimental groups. (**C**) PAS staining for the evaluation of goblet cell number in lung sections at 20× magnification (black arrows). (**D**) Bar graph showing percentage of goblet cells. Data are mean ± S.E.M. n = 10 mice per group. #### *p*< 0.0001 vs. naïve; ** *p* = 0.0085 and **** *p* < 0.0001 vs. vehicle.

**Figure 4 ijms-23-13300-f004:**
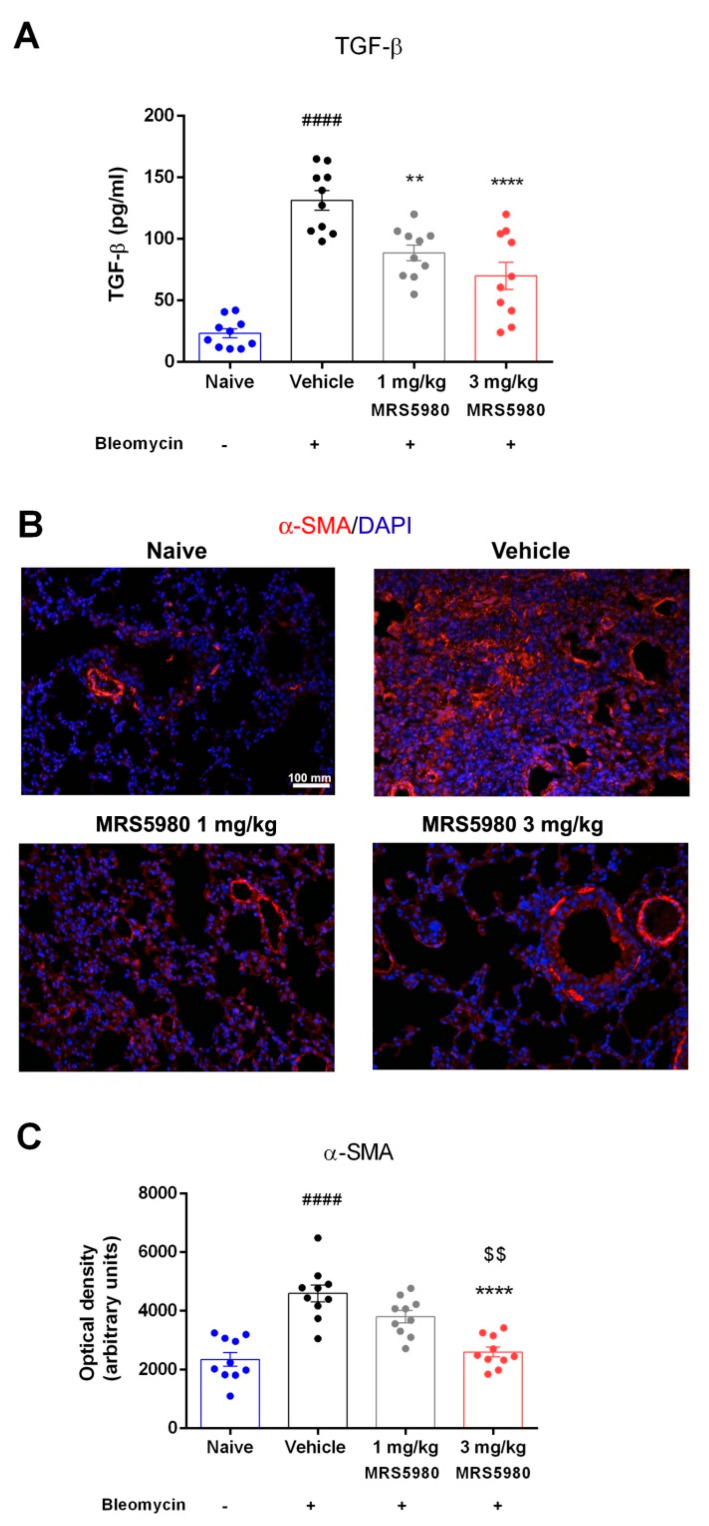
(**A**) Assessment of TGF-β production, a pro-fibrotic marker. Values are expressed as pg/µL. (**B**) Determination of fibroblast differentiation into myofibroblast. Lung tissue sections were labelled by immunofluorescence staining with α-smooth muscle actin (α-SMA) antibody (red) and DAPI (blue) to counterstain the nuclei (magnification 20×). Images show the inhibition of α-SMA expression, a marker of the transformation of fibroblasts into myofibroblasts. (**C**) Densitometric data are reported as relative optical density (OD) in each experimental group (naïve 2352 ± 231 OD, vehicle 4599 ± 287 OD, 1 mg/kg 3808 ± 204 OD and 3 mg/kg 2064 ± 168 OD). Data are mean ± S.E.M., n = 10 animals per group. #### *p* < 0.0001 vs. naïve, ** *p* < 0.01 **** *p* < 0.0001 vs. vehicle and $$ *p* < 0.01 vs. 1 mg/kg.

**Figure 5 ijms-23-13300-f005:**
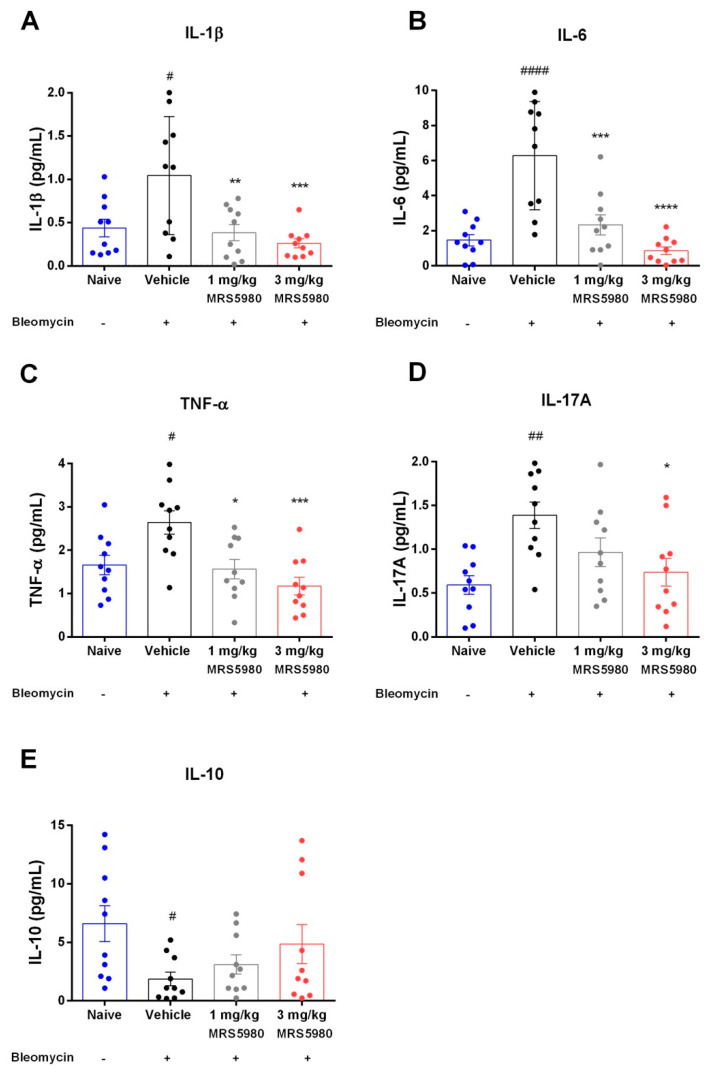
Determination of inflammatory markers. Analysis of IL-1β (**A**), IL-6 (**B**), TNF-α (**C**), IL-17A content (**D**) and IL-10 (**E**) in plasma samples of different experimental groups. The values are expressed as pg/mL of plasma. Data are mean ± S.E.M., n = 10 animals per group. # *p* < 0.05, ## *p* < 0.01 and #### *p* < 0.0001 vs. naïve; * *p* < 0.05, ** *p* < 0.01, *** *p* < 0.001 and **** *p* < 0.0001 vs. vehicle.

**Figure 6 ijms-23-13300-f006:**
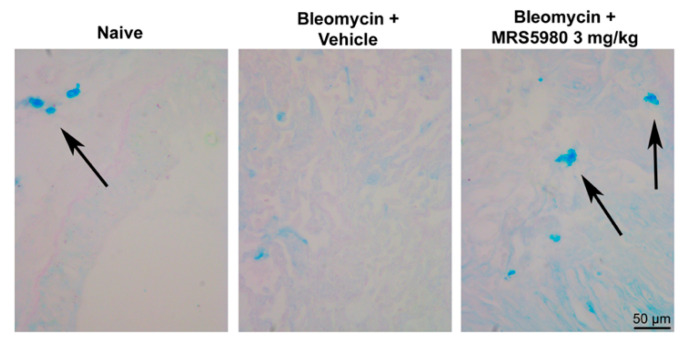
Determination of mast cell degranulation. Representative micrographs of Astra Blue-stained lung sections (magnification 40×) in naïve animals and animals treated with bleomycin plus vehicle or MRS5980, 3 mg/kg. Black arrows indicate mast cells.

**Figure 7 ijms-23-13300-f007:**
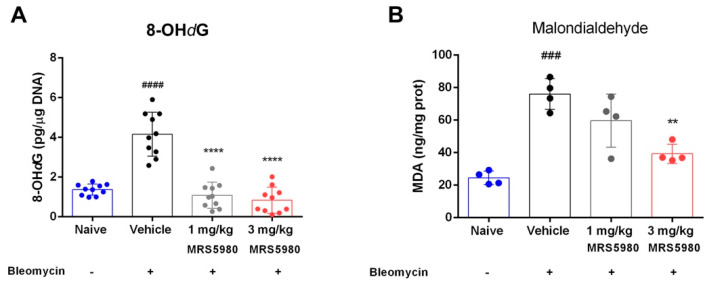
Evaluation of oxidative stress markers. Levels of 8-OH*d*G (**A**), a marker of DNA damage induced by production of free radicals. n = 10 animals per group. Levels of MDA (**B**), a marker of end-products of cell membrane lipid peroxidation by ROS, are evaluated in lung tissues. n = 4 animals per group. Data are mean ± S.E.M. #### *p* < 0.0001 and ### *p* < 0.001 vs. naïve; **** *p* < 0.0001 and * *p* < 0.05 vs. vehicle.

## Data Availability

Data are available upon request.

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
