# Peer review of "Adenosine A3 Receptor (A3AR) Agonist for the Treatment of Bleomycin-Induced Lung Fibrosis in Mice"

_ijms, 2022, doi:10.3390/ijms232113300_

Round 1

Reviewer 1 Report

This research article is well-written and engaging to read. The novelty of the study brings a magnificent sight for further therapeutic options for pulmonary fibrosis. Here are a few questions/comments I would like to ask.

1. H&E stain is the standard morphometrical analysis method. However, there are other special staining methods such as Masson Trichrome, which stains collagen and muscles, or Movat Pentachrome, which stains multiple tissue components including college, muscle and mucin (goblet cells). It will be more direct and efficient to see the deposition and morphological changes. Can the authors explain the significance of the current methods used in this study versus others?

2. In the immunofluorescent analysis of α-SMA expression, the authors stated the α-SMA levels were reduced. Can the authors clarify whether they analysed the α-SMA in whole tissue expression or only airway expression or only parenchymal expression? In Figure 4B, we can clearly see the α-SMA expression in the airway, parenchyma and vessels. 

Author Response

The authors would like to thank the reviewer for the comments relating to our paper.

Here are our replies.

  1. In our study we preferred to use three different staining methods to better analyze different structures in the lung tissue. In fact, H&E was used to have a whole morphological evaluation of the lung parenchyma (Figure 1C) and to analyze the bronchial smooth muscle layer status (Figure 3A), to measure the thickness of the muscle layer. The Picrosirius red staining was used in Figure 2B to evaluate the collagen deposition in interstitial lung spaces; finally, the PAS-stained preparations were used to evaluate bronchial mucosa goblet cell hyperplasia (Figure 3C), in order to calculate the percentage of PAS-positive goblet cells over total bronchial epithelial cells.
  2. In Figure 4B-C, we analyzed the expression of α-SMA in the whole tissue, taking care to exclude the perivascular and peribronchial muscular layers. This immunofluorescent analysis was performed to evaluate the activation of fibroblasts into myofibroblasts; this trans-differentiation induces a massive deposition of collagen in the lung tissue. This actin isoform is also abundant in vascular smooth muscle cells surrounding the vessels and the bronchi, for this reason we exclude perivascular and peribronchial muscular layers.

Reviewer 2 Report

                The present manuscript addresses the role of MRS5980, an A3AR agonist, in a murine model of lung fibrosis. The results showing the protective effect of the agonist are clear and credible. However, some conclusions reached must be toned down and/or consolidate since the degree of experimental development is limited.   1) More tissue markers of fibrosis should be added to the study (collagen, vimentin...) to consolidate this conclusion.   2) The conclusion reached for changes in IL-10 levels is unclear. The changes between vehicle vs. treatment are not significant in contrast to IL-6, THFalpha... The measurement has a lot of variability.   3) According to Morschi et al. 2008 A3AR receptors have a large effect on eosinophil recruitment. This aspect should be evaluated. Furthermore, the effect of A3AR on the activation of different leukocyte populations in the lung should be studied to increase the impact of the manuscript.   4) To conclude that MRS5980 has an effect on oxidative stress, more evidence must be provided. Therefore, Figure 7 must be expanded to make a robust study of the effect of the agonist on oxidative parameters.

Author Response

The authors would like to thank the reviewer for the comments relating to our paper.

Here are our replies.

  1. Many markers of fibrosis are really important to demonstrate the fibrosis in the lung tissue. However, in this study there are several reliable markers demonstrating the lung fibrosis, according to several publications [2, 28,29, 32,33,35,39].

In fact, we demonstrated the presence of fibrosis with different methods:

  1. Functional assay of fibrosis through the analysis of the Pressure at Airway Opening (PAO) (Figure 1B), a measurement of airway resistance to inflation, a functional parameter indicative of fibrosis-induced lung stiffness.
  2. The hydroxyproline content in lung tissue homogenates (Figure 2A). In animals, hydroxyproline is found almost entirely in collagen, and its content in tissue hydrolysates is a direct measure of the amount of collagen present, hence proportionally to fibrosis.
  3. Histopathological evaluation of collagen deposition was performed by Picrosirius red staining (Figure 2B). This staining is used to evaluate the collagen deposition in interstitial lung spaces to evaluate the fibrosis level in alveolar structure.
  4. TGF-β signaling pathway assessment and evaluation of fibroblasts activation by the expression of alpha-SMA (Figure 4 A-C). An increased TGF-β expression contributes to the formation and expansion of pulmonary fibrosis [18]. The bleomycin administration produced a large increase of TGF-β levels in plasma samples and the treatment with MRS5980 reduced this increase. It is well known that TGF-β signaling regulates the expression of α-SMA, a marker of fibroblast activation and myofibroblast differentiation [6]. We evaluated the expression of α-SMA in lungs with an immunofluorescence analysis, highlighting a large increase of these levels in the vehicle group; while in MRS5980-treated animals, α-SMA levels were considerably reduced.
  5. Finally, the selected model of bleomycin induced lung fibrosis in mice is a well characterized murine model in use today for lung fibrosis and the development of fibrosis induced by the injection of bleomycin is widely documented in the literature [2,5,26].

  1. In Figure 5E, the levels of the regulatory cytokine IL-10 are significantly decreased in the vehicle group in comparison to the naïve group, and the treatment with compound MRS5980 suggests an increase in these levels in a dose-dependent manner even without reaching the statistical significance. As suggested by the Reviewer, we changed the text in the manuscript with the following sentence in the Result Section: “Moreover, bleomycin treatment significantly decreased the levels of the regulatory cytokine IL-10 (from 6.5 ± 2.3 to 1.7 ± 0.7 pg/ml), while MRS5980 administration showed a trend toward increased IL-10 levels in a dose-dependent manner (3.1 ± 1.0 and 4.7 ± 3.0 pg/ml, respectively), even if it was not statistically significant Figure 5E).”, and in the Discussion Section “Moreover, MRS5980 treatment tended to increase IL-10 levels suggesting an involvement of this regulatory cytokine.”
  2. We agree with the reviewer that the aspect of eosinophil recruitment should be evaluated. This would have been an interesting parameter but unfortunately, we did not collect BAL sample. We tried to highlight the eosinophil cationic protein (ECP) in the lung specimen with an immunofluorescent staining analysis (as reported in Bystrom, J., Amin, K. & Bishop-Bailey, D. Analysing the eosinophil cationic protein - a clue to the function of the eosinophil granulocyte. Respir Res 12, 10 (2011). https://doi.org/10.1186/1465-9921-12-10 and Mahmudi-Azer S et al., J Allergy Clin Immunol, 105 (6) part 1, doi:10.1067/mai.2000.106930); unfortunately, the obtained results were not convincing probably because the antibody did not work properly. It produced a lot of noise background likely because of its lower specificity.
  3. We thank the reviewer for suggesting to evaluate different leukocyte population, but in this study we did not collect PMNs and so we have planned to investigate this aspect in the future.
  4. In order to corroborate our results on oxidative stress, we performed MDA determination on the remaining lung tissues (n=4 per group). We added in the text the following paragraph: “To confirm the results on oxidative stress effects, we evaluated the production of MDA, as TBARS, end-products of cell membrane lipid peroxidation by ROS, a reliable marker of oxidative tissue injury (Figure 7B). The production of TBARS (MDA equivalents) was markedly increased in the vehicle group in comparison to naïve animals (from 24.47 ± 2.07 to 76.00 ± 4.70 ng/mg of protein). As shown in Figure 7B, MRS5980 treatment significantly reduced TBARS production at the highest dose (1mg/kg 59.61 ± 8.18 and 3 mg/kg 39.30 ± 2.95 ng/mg of protein).”.
